# Interplay between Senescence and Macrophages in Diabetic Cardiomyopathy: A Review of the Potential Role of GDF-15 and Klotho

**DOI:** 10.3390/biomedicines12040759

**Published:** 2024-03-29

**Authors:** Ghada M. Almohaimeed, Asma S. Alonazi, Anfal F. Bin Dayel, Tahani K. Alshammari, Hanan K. Alghibiwi, Maha A. Alamin, Ahmad R. Almotairi, Nouf M. Alrasheed

**Affiliations:** 1Department of Pharmacology and Toxicology, College of Pharmacy, King Saud University, Riyadh 11451, Saudi Arabia; 443203543@student.ksu.edu.sa (G.M.A.); aaloneazi@ksu.edu.sa (A.S.A.); abindayel@ksu.edu.sa (A.F.B.D.); talshammary@ksu.edu.sa (T.K.A.); halghibiwi@ksu.edu.sa (H.K.A.); mahaali@ksu.edu.sa (M.A.A.); 2Department of Pathology, College of Medicine, King Saud University, Riyadh 11461, Saudi Arabia; aalmotairi1@ksu.edu.sa

**Keywords:** diabetic cardiomyopathy, macrophage, Klotho, GDF-15, senescence

## Abstract

Type 2 diabetes mellitus (T2DM) is a critical health problem, with 700 million diagnoses expected worldwide by 2045. Uncontrolled high blood glucose levels can lead to serious complications, including diabetic cardiomyopathy (DCM). Diabetes induces cardiovascular aging and inflammation, increasing cardiomyopathy risk. DCM is characterized by structural and functional abnormalities in the heart. Growing evidence suggests that cellular senescence and macrophage-mediated inflammation participate in the pathogenesis and progression of DCM. Evidence indicates that growth differentiation factor-15 (GDF-15), a protein that belongs to the transforming growth factor-beta (TGF-β) superfamily, is associated with age-related diseases and exerts an anti-inflammatory role in various disease models. Although further evidence suggests that GDF-15 can preserve Klotho, a transmembrane antiaging protein, emerging research has elucidated the potential involvement of GDF-15 and Klotho in the interplay between macrophages-induced inflammation and cellular senescence in the context of DCM. This review explores the intricate relationship between senescence and macrophages in DCM while highlighting the possible contributions of GDF-15 and Klotho.

## 1. Introduction

Diabetes mellitus is a metabolic condition defined by elevated blood glucose levels resulting from impaired insulin production or action. Type 2 diabetes mellitus (T2DM) is a critical health issue that greatly impacts human quality of life and healthcare costs [1]. Over the past four decades, the prevalence of T2DM has increased globally, particularly in countries undergoing rapid epidemiological transitions, such as those in Asia, the Middle East, and North Africa. In 2019, approximately 6.0% of men and 5.0% of women were affected by T2DM, according to Global Burden of Disease data, representing a substantial increase from 3.9% in men and 3.5% in women in 1990 [2].

According to the International Diabetes Federation, 537 million adults worldwide had diabetes in 2021, which is expected to rise to 643 million and 783 million by 2030 and 2045, respectively. In the Middle East, one in six adults (73 million) had diabetes in 2021, which is predicted to increase to 95 million by 2030 and 136 million by 2045. Diabetes caused 796,000 deaths in 2021, notably while one in three adults living with diabetes was undiagnosed [3]. Various risk factors contribute to T2DM, including genetics, lifestyle, and environmental factors, such as physical activity and obesity. Dietary factors, such as the consumption of red and processed meat, sugary beverages, refined grains, and low-quality carbohydrates, contribute greatly to the T2DM epidemic. The risk of T2DM is also determined by age, family history, and ethnic background [4].

Individuals with uncontrolled and untreated elevated blood glucose levels may develop various cardiovascular complications, including blood clotting, diabetic cardiomyopathy (DCM), and hypertension. These complications can potentially lead to severe outcomes, such as diabetes-induced heart failure (HF) [5]. Notably, diabetes is one of the seven major risk factors for cardiovascular disease (CVD), according to the American Heart Association [6]. Indeed, diabetic patients face a higher risk of HF than individuals of the same age without diabetes: According to the Framingham Heart Study conducted by the University of Massachusetts Medical School, men and women with diabetes had a two- to five-times greater chance of developing HF than those without diabetes [7,8]. Furthermore, CVD, particularly coronary artery disease and ischemic cardiomyopathy, is the leading cause of death in diabetic patients [9]. Elevated blood glucose levels damage the small blood vessels in the heart muscle, impairing its ability to contract and pump blood. This chronic impairment can weaken heart muscles, thereby establishing DCM, and exacerbate the symptoms of diastolic and systolic cardiac failure [8]. Therefore, DCM represents a microvascular complication characterized by structural and functional abnormalities in the heart muscle. These abnormalities encompass left ventricular hypertrophy, fibrosis, lipid accumulation within cardiomyocytes, and contractile and diastolic dysfunction [10]. Shirley Rubler first identified DCM in four patients with diabetic glomerulosclerosis and HF in 1972. These patients exhibited myocardial hypertrophy, fibrosis, and cardiac dysfunction in the absence of coronary artery disease, hypertension, and valvular disease [11], consistent with the European Society of Cardiology’s definition of cardiomyopathy [12].

Numerous etiological factors contributing to the development and progression of DCM have been identified, including hyperglycemia, hyperinsulinemia, insulin resistance, mitochondrial dysfunction, altered calcium homeostasis, increased production and accumulation of advanced glycation end products (AGEs) in cardiac cells as a consequence of prolonged hyperglycemia, and inflammation [13]. Senescence, a state of irreversible cell cycle arrest, and macrophage-mediated inflammation have emerged as key processes in the development and progression of diabetic cardiomyopathy [14,15]. Recent studies have implicated growth differentiation factor-15 (GDF-15) and Klotho, two factors involved in stress responses and aging, as potential regulators in this complex interplay of processes [16,17] (Figure 1). However, the interplay between these factors and processes in the development of DCM remains largely unclear but retards the development of effective clinical treatments. Accordingly, this review aims to provide a concise overview of the latest findings in this field, elucidate the potential roles and interactions of relevant processes and molecular markers, and identify essential gaps in knowledge that inhibit progress in pre-clinical and clinical research.

## 2. Pathophysiological Mechanisms of DCM

The pathophysiological mechanisms of DCM are multifactorial, encompassing metabolic alteration, AGE accumulation, mitochondrial dysfunction, lipotoxicity, glucotoxicity, inflammation, and oxidative stress (Figure 2) [19,20,21].

Persistent hyperglycemia in diabetic patients leads to metabolic dysregulation in cardiomyocytes, including insulin resistance, increased fatty acid (FA) uptake, and impaired glucose utilization. Typically, 70% of myocardial energy comes from FA oxidation, with the remaining 30% derived from glucose metabolism through mitochondrial oxidative phosphorylation [22]. Insulin-impaired action on adipose tissue and the liver contributes to increased lipolysis, thereby elevating blood FA contents and promoting the overexpression of a cluster of differentiation 36, essential for FA uptake, eventually leading to increased FA oxidation. Additionally, insulin resistance causes hyperinsulinemia and impaired insulin signaling, inhibiting glucose transporter 4-mediated glucose uptake. This triggers the overproduction of reactive oxygen species (ROS) in the mitochondrial respiratory chain, reducing myocardial contractility and promoting myocardial fibrosis—a hallmark of DCM. Elevated ROS production leads to oxidative stress and myocyte inflammation, causing lipid accumulation and lipotoxicity [23,24]. Excessive FA can metabolize through non-oxidative pathways, forming lipotoxic intermediates, such as ceramides and diacylglycerol. These intermediates lead to endoplasmic reticulum stress and cardiomyocyte apoptosis, ultimately progressing to DCM. Prolonged hyperglycemia promotes AGE formation and protein kinase-C activation, which initiates oxidative stress by stimulating ROS production, ultimately causing oxidative damage to cardiomyocytes and impaired myocardial contractility [25,26]. A significant increase in AGEs triggers nuclear factor kappa-B (NF-κB) signaling pathways, thereby inducing inflammation, cytokine and chemokine production, and profibrotic factor activation, such as matrix metalloproteinase (MMP) and transforming growth factor-beta (TGF-β), as well as myocyte apoptosis. These pathways can induce functional and structural damage, leading to cardiomyocyte death, left ventricular remodeling, and systolic dysfunction [23]. Elevated FA accumulation activates the peroxisome proliferator-activated receptors-α and -γ (PPARα/PPARγ) cofactor-1α pathway, enhancing the transcription of genes for FA uptake and oxidation regulation. The overexpression of PPARα in the heart increases FA absorption and oxidation, leading to reduced mitochondrial function, the loss of metabolic flexibility, and ROS production by the mitochondria [21]. In diabetes mellitus, coronary microvascular dysfunction is an important characteristic of diabetes mellitus-related complications. It is linked to various other risk factors that significantly affect cardiovascular disease morbidity and mortality. Chronic uncontrolled high blood glucose contributes significantly to the dysfunction of microcirculation as a common chronic feature. The pathogenesis of this microvascular complication is complex and not completely understood [27]. However, hyperglycemia contributes to the activation of various signaling pathways at microvascular levels, resulting in the remodeling of the microvessels as well as myocardial tissue, with possible impairment of the microvascular supply to the myocardium [28]. Microvascular remodeling includes the thickening of the arteriole wall, the narrowing of the lumen, fibrosis, and capillary rarefaction. Similarly, hypertension is another risk factor that leads to coronary microvascular alterations and promotes the progression of endothelial dysfunction and capillary rarefaction [29]. Hypertension-induced changes in myocardial morphology and function also contribute to left ventricular hypertrophy and myocardial fibrosis [29]; indeed, hypertension and type 2 diabetes are common comorbidities. Accordingly, uncontrolled blood glucose levels and blood pressure are interlinked because of diabetes-related or hypertension-related cardiovascular complications associated primarily with microvascular function deterioration. Both diseases can stimulate a mutual molecular mechanism, such as oxidative stress, inflammatory and immune system activation, and others. All of these are linked to microvascular dysfunctions that possibly contribute to the relationship between diabetes, hypertension, and coronary microvascular disorders [30]. Furthermore, overactivated macrophages contribute to oxidative stress and mitochondrial damage through ROS overexpression, resulting in progressive cardiac dysfunction and cardiomyopathy.

## 3. Macrophages and Inflammation in DCM

Inflammatory processes significantly contribute to the pathogenesis of DCM [31]. T2DM induces chronic low-grade inflammation in the heart tissue, leading to severe structural and functional alterations [32]. Elevated glucose levels lead to the release of pro-inflammatory cytokines (interleukin (IL)-6, IL-18, pro-IL-1ß, and tumor necrosis factor-alpha (TNF-α) [25,33,34]. Recent studies have shown that the STZ-induced T2DM animal model exhibits fibrosis and tissue damage, features of chronic inflammation [32,35]. Moreover, both in vitro and animal research have revealed that elevated blood glucose levels are a substantial factor in diabetes-induced myocardial fibrosis. Moreover, when cardiac fibroblasts are cultivated in an environment with a high concentration of glucose, the cells tend to produce an excessive quantity of ECM proteins, including collagen, fibronectin, and matricellular macromolecules [36,37]. The development of diabetic cardiomyopathy using the STZ-induced T2DM animal model was associated with an increase in cardiac immune cell invasion, specifically T lymphocytes and macrophages, indicating cardiac chronic inflammation [20,38,39,40]. These findings suggest the presence of chronic inflammation in the myocardial interstitium, potentially leading to cardiac dysfunction and an increased risk of cardiac events, such as DCM.

Macrophages, pivotal immune cells, play multiple roles in the pathogenesis and progression of DCM. Reflecting their remarkable plasticity, macrophages can undergo polarization (a phenotypic shift) in response to microenvironmental cues, with two distinct activation states: M1 and M2. M1 macrophages promote inflammation and pro-inflammatory cytokine production, while M2 macrophages participate in tissue repair, exhibiting anti-inflammatory properties [23,31].

DCM is associated with an imbalanced M1-to-M2-macrophage ratio [15]. Multiple studies have reported a predominantly M1 phenotype in DCM, contributing to cardiac fibrosis and impaired cardiac function [15,41]. M1 macrophage-activated pro-inflammatory cytokines, such as tumor necrosis factor-alpha (TNF-α), interleukin-1 beta (IL-1β), and IL-6, promote inflammation, oxidative stress, and cardiomyocyte apoptosis. Additionally, M1 macrophages intensify the inflammatory response by recruiting other immune cells. Conversely, the lower levels of M2 macrophages, which regulate tissue repair by releasing anti-inflammatory cytokines, such as IL-10 and TGF-β, and promote tissue remodeling and angiogenesis, may impede cardiac healing mechanisms and exacerbate the progression of DCM [42,43].

Several factors contribute to the dysregulation of macrophage polarization in DCM. Chronic hyperglycemia and insulin resistance in diabetic patients initiate an inflammatory response marked by increased pro-inflammatory cytokine production [25]. Macrophages infiltrate cardiac tissue in response to chemokine activation, propagating the inflammatory response. Additionally, hyperglycemia and insulin resistance contribute to macrophage polarization toward the pro-inflammatory M1 phenotype [41]. Hyperglycemia induces oxidative stress through increased ROS production, which activates various signaling pathways involved in M1 polarization [43]. Diabetes-associated metabolic disturbances, such as dyslipidemia and AGEs, further contribute to macrophage polarization. Dyslipidemia promotes the production of pro-inflammatory lipid mediators, such as oxidized low-density lipoprotein, leading to M1 polarization. AGEs, formed by non-enzymatic protein glycation, activate receptors that promote pro-inflammatory responses, thereby impairing M2 macrophage polarization and function [15]. Lipids have a substantial effect on macrophage polarization and influence macrophage functions. Previous studies have shown that oxidized low-density lipoproteins (oxLDLs) induced M1 polarization, thereby contributing to the inflammatory process of atherosclerosis [44,45]. For example, oxLDL induced the production of pro-inflammatory cytokines, the expression of HLA-DR and CD86, and T-cell proliferation [46]. Another study reported that the exposure of monocytes to a low concentration of oxLDL induced epigenetic histone modifications that resulted in a long-lasting proatherogenic macrophage phenotype characterized by increased pro-inflammatory cytokine production and foam cell formation [47]. Peroxisome proliferator-activated receptors (PPARs) and liver X receptors (LXRs) are other well-known lipid-binding factors with known effects on macrophage polarization [48]. Additionally, senescent cardiomyocytes can promote M1-biased macrophage polarization by releasing senescence-associated secretory phenotype (SASP) components [49].

## 4. Senescence and DCM

Cellular senescence refers to a state of irreversible growth arrest characterized by changes in cell morphology, gene expression, and the secretion of pro-inflammatory SASP molecules—enriched in pro-inflammatory cytokines, growth factors, and profibrotic proteins [49]. Leonard Hayflick and Paul Moorhead discovered cell senescence in 1961 by observing that human fibroblast cells stopped proliferating in vitro after 40 to 60 passages [50]. Senescence was later described as irreversible cell cycle arrest in the G1 phase, a non-proliferative but viable state [51]. Cellular senescence may be harmful or advantageous, depending on the biological setting and timing. SASP molecules induce a local inflammatory response that promotes the phagocytosis of senescent cells and the remodeling of damaged tissue [52]. However, chronic senescence-mediated inflammation can lead to abnormal tissue remodeling and fibrosis [53].

Various factors induce cellular senescence, including oxidative stress, DNA damage, telomere dysfunction, and chronic inflammation. Cellular senescence is considered a hallmark of aging since it reduces the capacities of numerous cell pools, including progenitor and stem cells, needed to replace damaged tissue [51]. Evidence indicates that T2DM-induced pathological changes stimulate cellular senescence [54,55]. Various studies in diabetic adults have also reported a strong association between T2DM and cardiovascular or early vascular aging [14,56,57]. Diabetes affects cardiac stem cells, inhibiting their reparative potential and triggering cellular senescence and the SASP, irrespective of biological aging [14].

In diabetes, hyperglycemia and insulin resistance are significant drivers of cell senescence, including cardiomyocytes [58]. Senescence mechanisms in diabetic hearts include telomere dysfunction and attrition caused by increased oxidative stress and chronic hyperglycemia [59]. Dysfunctional telomeres activate the p53–p21 pathway, leading to cell cycle arrest and cellular senescence. This causes phenotypic changes in cardiomyocytes, resulting in an abnormal SASP enriched in pro-inflammatory and profibrotic substances, leading to the development of DCM [59]. The kinases adenosine monophosphate-activated protein kinase (AMPK) and mammalian target of rapamycin (mTOR) participate in various senescence mechanisms. In DCM, impaired AMPK activity and enhanced mTOR signaling disrupt cellular homeostasis, promoting cell cycle progression and premature senescence [60,61]. Inhibiting mTOR with rapamycin reduces cellular senescence in various cultured cells [61]. The transcription factor NF-κB also regulates inflammation, apoptosis, and senescence [62]. NF-κB signaling is activated in response to pro-inflammatory cytokines, including IL-1β and TNF-α, promoting cellular senescence through the upregulation of cyclin-dependent kinase inhibitors, such as p16^INK4a^ and p21, and the secretion of SASP factors [62]. This can cause chronic low-grade inflammation or inflammaging—a characteristic feature of DCM.

Increased oxidative stress, a hallmark of DCM, promotes cellular senescence [63,64]. Oxidative stress-induced ROS generation can directly damage DNA, activate p53, and upregulate cyclin-dependent kinase (CDK) inhibitor expression, leading to cell cycle arrest and senescence [60]. ROS can also activate various protein kinases, including the mitogen-activated protein kinase and protein kinase-C pathways, further exacerbating senescence-promoting signaling [65]. Senescent cells further secrete extracellular matrix (ECM) proteins that promote fibrosis. In DCM, senescent cardiomyocytes and other cells contribute to cardiac fibrosis, leading to heart stiffness and impaired ventricular function [66]. Additionally, senescent cardiomyocytes can reduce regenerative capacity and cannot adequately replace damaged cells, resulting in cardiac function decline and compromised tissue repair in DCM. Senescent cardiomyocytes exhibit disrupted metabolism, including impaired mitochondrial function and altered stress response pathways, further contributing to cardiac dysfunction and increasing susceptibility to cellular damage [14].

## 5. Relationship between Macrophages and Senescence in DCM

The interplay between senescence and macrophages is a critical mechanism underlying DCM progression. Under hyperglycemic conditions, cardiomyocytes in diabetic patients undergo stress-induced premature senescence. Senescent cardiomyocytes release inflammatory cytokines that recruit monocytes, which differentiate into macrophages and infiltrate the myocardium. Senescent cardiomyocytes can directly modulate macrophage polarization toward the M1 phenotype by releasing SASP components. These macrophages sustain the senescence phenotype by amplifying the pro-inflammatory response [49]. Notably, macrophages regulated cellular senescence by influencing their microenvironment in a p16^INK4a/Luc^ mouse model [67].

M1 macrophages promote senescence by secreting inflammatory cytokines, such as TNF-α and IL-6 [49,68]. The presence of senescent cells and macrophages further contributes to cardiac remodeling, fibrosis, and impaired contractility. The interplay between senescence and macrophages facilitates cardiac fibrosis, as senescent cells produce MMPs that degrade the ECM, while macrophages enhance fibrotic tissue deposition [49]. In support of this, chemotherapy in breast cancer cells with a mutated *TP53* gene promotes senescent cells and cellular survival via phagocytosis and macrophages [69], indicating a functional correlation between senescence and macrophages.

In a streptozotocin-diabetic mouse model, pathological features of cardiomyopathy were attenuated by the in vivo administration of Klotho for a period of 12 weeks. In addition, Klotho was found to reduce inflammatory and stress-related features in H9C2 cardiomyoblasts exposed to high levels of glucose [70]. In individuals with acute heart failure, the soluble alpha-Klotho circulatory level was significantly correlated to patients’ responsiveness to intensive treatment [71]. In line with this, the genetic mutation of *Klotho* accelerated aging in mice in different physiological systems and organs, including cardiac muscle [72]. A previous review showed that the preanalytic characteristics of GDF-15 in clinical settings indicate that a reduced level of circulatory GDF-15 is linked to longevity [73]. In support of this, the level of GDF-15 in aged and healthy individuals was found to be correlated to age. Furthermore, in chronic heart failure patients, the GDF-15 serum level was elevated and associated with the severity of the disease [74].

In cardiomyocytes, the functional alterations of replicative senescence are associated with chronic inflammation and cell death [75]. In a study that recruited twenty heart failure patients, it was found that compared to normal controls, inflammation contributes to the extracellular matrix of the endomyocardium and affects cardiac remodeling [76]. A proteomic study conducted on serum samples isolated from cardiomyopathy patients has shown that dysregulation in extracellular matrix protein is associated with DCM [77]. In addition, pathological changes in myocardial remodeling were found to be modulated by senescent cardiomyocytes [78]. Additionally, the genetic profiling of macrophages was found to be correlated with changes associated with myocardial infarction healing and modeling [79]. Immunohistochemical studies conducted on biopsy samples isolated from DCM patients have shown significant immunoreactivity of macrophage markers, including whole and M2 macrophages [80]. This evidence supports the cross-talk between senescence and macrophages and their contribution to DCM and cardiac remodeling as we age.

## 6. Role of Klotho in DCM

Klotho, a transmembrane protein with an intracellular tail and extracellular domain, belongs to the mitochondrial protein family (Figure 3). Discovered in 1997 by Kuro-o et al. [72] as an aging suppressor gene, Klotho exists in two forms: α-Klotho and β-Klotho. While α-Klotho is expressed in various organs, including the kidney, parathyroid gland, brain, adipose tissue, small intestine, and heart, β-Klotho is found in adipose tissue, the pancreas, and liver. Both forms may have their extracellular domains cleaved, releasing soluble Klotho into the blood, cerebrospinal fluid, and urine [81].

Klotho plays a protective role against CVDs by maintaining proper cardiac and vascular function [17]. It is implicated in defense mechanisms against heart hypertrophy and remodeling [17] and exhibits a protective effect in diabetic heart tissue by reducing oxidative stress [82]. Klotho exerts anti-inflammatory effects by inhibiting the expression of pro-inflammatory cytokines, such as IL-6 and TNF-α, thereby preserving the structural and functional integrity of the diabetic heart. Additionally, Klotho regulates the expression of profibrotic factors, such as TGF-β and connective tissue growth factor, thereby inhibiting fibrosis. It also promotes MMP activation, which degrades collagen and prevents fibrotic remodeling [82,83].

Klotho inhibits oxidative stress by reducing ROS production and enhancing antioxidant enzyme activity, thereby preventing apoptosis [84,85,86]. In a mouse model of T2DM, Klotho overexpression alleviated insulin sensitivity and metabolic disruption, ultimately attenuating DCM [87]. Furthermore, increased Klotho protein expression and function in renal tubular cells induced by GDF-15 suggest additional DCM-specific protective mechanisms [88]. In a recent study by Donate-Correa et al., in adults with T2DM and preserved kidney function, reduced Klotho levels were observed. Furthermore, reduced Klotho levels were associated with increased levels of inflammatory markers and a higher incidence of vascular disease and subclinical atherosclerosis [89]. This suggests that Klotho may be involved in the development and progression of diabetes-related vascular complications. Lower levels of Klotho were also found in adults with a higher risk of cardiovascular disease, such as obesity, smoking, diabetes, and higher levels of total cholesterol and triglycerides [90]. In both in vitro and in vivo experiments, Klotho treatment effectively suppressed high glucose-induced inflammation, ROS generation, and cardiac cell death, leading to improved cardiac function [91]. The antioxidative, anti-inflammatory, antiaging, and antiapoptotic effects of Klotho highlight its potential as a novel protective protein against DCM.

**Figure 3 biomedicines-12-00759-f003:**
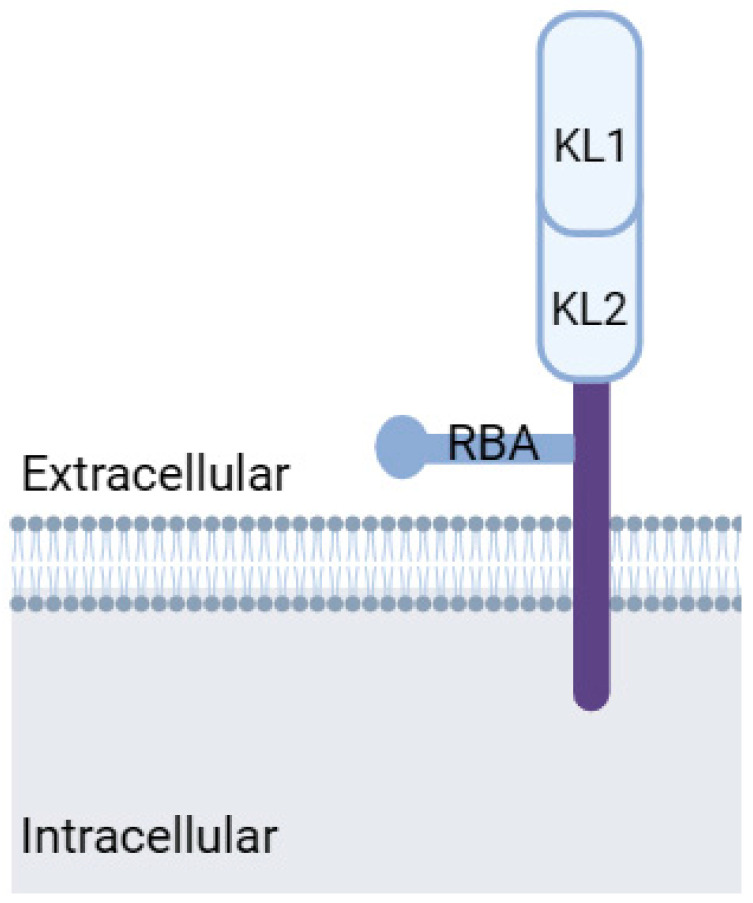
Klotho protein structure. KL1: Klotho domain 1, KL2: Klotho domain 2, RBA: receptor-binding arm. Modified from Kuro-o, 2019 [83]. Created with biorender.com.

## 7. Role of GDF-15 in DCM

GDF-15, a member of the TGF-β family, acts as a stress-responsive cytokine and is expressed by macrophages, epithelial cells, and adipocytes. It exerts anti-inflammatory effects by inhibiting macrophage activation, enhancing insulin sensitivity, and preventing diabetic complications [88]. Under normal conditions, human serum GDF-15 levels are low but are significantly elevated in various diseases, including malignancies, CVDs, renal disorders, and diabetes [92]. GDF-15 levels are notably high during pregnancy and can further be exacerbated by smoking, psychological states, and environmental stressors, making GDF-15 an attractive candidate for diagnostic and prognostic biomarkers, including in all-cause mortality [92,93,94]. Glial-derived neurotrophic factor family receptor α-like (GFRAL) is the receptor for GDF-15 and is only detected in the brain. Upon binding to GFRAL, GDF-15 promotes the activation of the coreceptor tyrosine kinase RET. This GDF-15/GFRAL/RET complex plays a critical role in weight regulation, as indicated by the promotion of weight loss in obese mice by recombinant GDF-15, which triggered a reduction in food intake. This anti-obesity action of GDF-15 was abrogated in GFRAL gene-deleted mice [95,96], while diet-induced obesity was exacerbated in GFRAL-deficient mice [96]. Remarkably, transgenic mice that overexpressed GDF-15 exhibited enhanced insulin sensitivity and glucose tolerance along with a marked reduction in body weight [95,97]. These beneficial metabolic effects provide new insights into GDF-15 as a potential target for diabetes treatment. Moreover, GDF-15 expression has a protective effect, as the absence of GDF15 is associated with increased damage in various tissues. For example, GDF-15 knockout mice demonstrated cardiac rupture after myocardial infarction [98]. It is noteworthy that GDF-15 plays a vital role in mitigating the inflammatory environment in diabetic hearts by suppressing the release of pro-inflammatory cytokines [16], regulating mitochondrial function, and reducing oxidative stress—the crucial factors driving DCM progression [99]. In DCM, excessive deposition of ECM proteins, such as collagen, leads to myocardial fibrosis. GDF-15 inhibits collagen synthesis, prevents fibrotic remodeling, and activates MMPs, which degrade and remodel the ECM, preventing excessive fibrosis [99]. Furthermore, GDF-15 exhibits potent antiapoptotic effects in the myocardium, thereby inhibiting cell death and preserving cardiac function. It has been observed that GDF-15 is induced in response to conditions that promote hypertrophy and dilated cardiomyopathy. Transgenic mice with cardiac-specific overexpression of GDF15 showed partial resistance to pressure overload-induced hypertrophy [73]. It also activates the intracellular PI3K/Akt and ERK1/2 signaling pathways to promote cell survival and inhibit apoptosis in diabetic hearts [16,99].

Despite all the protective effects of GDF-15, clinical studies in patients with DCM have shown that elevated GDF-15 levels correlate with adverse cardiovascular outcomes, including HF, myocardial infarction, and mortality [99,100]. Furthermore, GDF-15 levels have been found to be positively associated with markers of poor glycemic control, such as higher HbA1c levels [16,101,102,103]. Additionally, elevated GDF-15 levels have been associated with an increased risk of diabetic complications, such as DCM, chronic kidney disease, and diabetic retinopathy [101,103,104]. Jurczyluk et al. found that GDF-15 levels were highly increased in cardiomyocytes following an ischemic event and post-reperfusion [105]. In addition, GDF-15 levels were higher in patients with diabetic nephropathy than in diabetic patients without diabetic nephropathy and were associated with impaired kidney function [106]. Similar patterns were observed in diabetic retinopathy patients [107]. Currently, there are limited studies highlighting the link between GDF-15 and Klotho protein in the context of DCM. In different disease models, only one study investigated the correlation between GDF-15 and Klotho protein in acute kidney injury and kidney fibrosis and found that GDF-15 and Klotho protein levels have a strong correlation. GDF-15 enhanced Klotho expression in healthy mice and cultured tubular cells [108], whereas Klotho expression was reduced in GDF-15-deficient mice and conserved after GDF-15 administration [108]. Furthermore, GDF-15 and Klotho protein are both involved in the development and progression of fibrosis, a hallmark of DCM [109,110]. One study found that GDF-15 expression and accumulation are increased in the extracellular matrix of idiopathic pulmonary fibrosis. The study suggested that increased expression and accumulation of GDF-15 in the extracellular matrix contribute to the fibrotic process in idiopathic pulmonary fibrosis. This finding highlights the potential role of GDF-15 in promoting fibrosis in the lungs [110]. One study used a mouse model of myocardial infarction and found that treatment with Klotho improved cardiac function and reduced cardiac fibrosis [111]. Another investigated the role of Klotho in the development of cardiac fibrosis in a long-term rat model resembling type 1 diabetes mellitus and found that serum levels of Klotho were reduced in diabetic rats, possibly promoting the fibrotic process [112]. The exact mechanisms underlying this relationship remain unclear, though two hypotheses have been proposed: Chronic exposure to high GDF-15 levels may desensitize GDF-15 receptors, affecting the cardiomyocyte response. The desensitization of GDF-15 receptors can compromise its anti-inflammatory effects, apoptosis regulation, oxidative stress modulation, and cell survival promotion, thus impairing its cardioprotective effects and potentially contributing to DCM progression [113]. Furthermore, impaired GDF-15 signaling due to receptor desensitization may disrupt the balance between profibrotic and anti-fibrotic factors, leading to adverse remodeling in the diabetic heart. Second, GDF-15 upregulation may be a compensatory mechanism to counterbalance the inflammatory state in DCM [114]. However, no study has specifically assessed the roles and interactions of GDF-15 and Klotho in DCM or how macrophages mediate this relationship.

## 8. Conclusions and Future Directions

This review explored the interplay between senescence and macrophages in diabetic cardiomyopathy, specifically focusing on the potential role of GDF-15 and Klotho. The interplay between senescence and macrophages in diabetic cardiomyopathy involves complex mechanisms associated with inflammation, fibrosis, and cardiac dysfunction. However, the underlying molecular mechanisms remain largely unclear. Further research is needed to elucidate the exact mechanisms by which GDF-15 and Klotho influence senescence and macrophage function in DCM, the mechanisms of GDF-15 receptor desensitization in DCM, and the impact of receptor desensitization on GDF-15 signaling and DCM pathogenesis. The review also highlighted the potential of other factors, such as alterations in ligand–receptor interactions or downstream signaling components, in the dysregulation of GDF-15 signaling in DCM. In addition, since DCM is characterized by structural and functional changes in the heart, the precise impact of the interplay between senescence and macrophages on cardiac remodeling is an important topic in future clinical research.

Although pre-clinical studies have elucidated some of the interplay between senescence and macrophages [113,115], further clinical studies are needed to validate these findings. Clinical trials investigating the diagnostic and prognostic value of GDF-15 and Klotho in DCM and exploring the therapeutic potential of targeting senescence and macrophages, including the modulation of GDF-15 and Klotho, are warranted to translate the research findings into clinical practice. Further studies are needed to determine how senescent cells and macrophages contribute to cardiac remodeling, such as fibrosis, hypertrophy, and angiogenesis, and to learn the specific effects of GDF-15 and Klotho on cardiac function in DCM. This should include studying their impact on cardiomyocyte senescence, macrophage polarization, and the overall cardiac remodeling process. Additionally, exploring the potential cross-talk between GDF-15, Klotho, and other signaling pathways implicated in DCM may provide valuable insights into disease development.

## Figures and Tables

**Figure 1 biomedicines-12-00759-f001:**
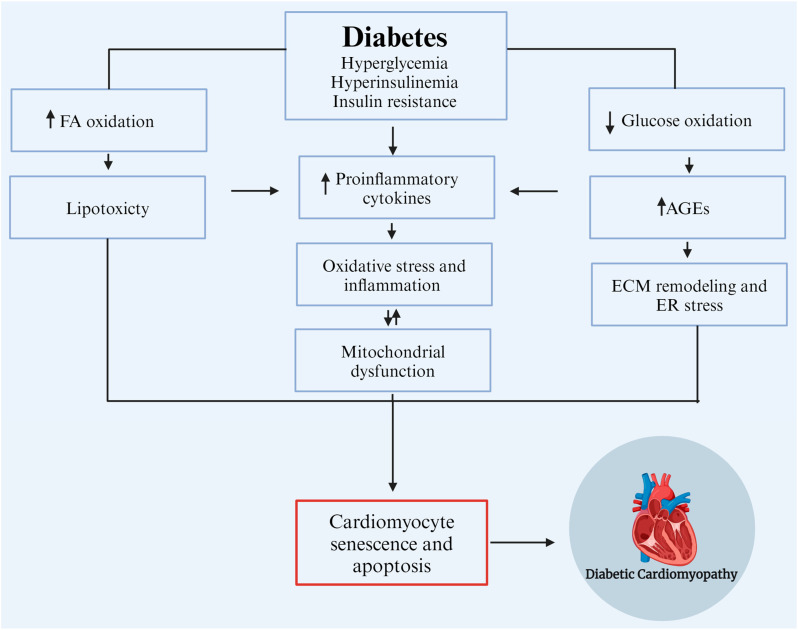
Etiological factors contribute to the development and progression of diabetic cardiomyopathy (DCM). AGEs: advanced glycation end products, ECM: extracellular matrix, ER: endoplasmic reticulum, and FA: fatty Acids. Modified from Borghetti et al., 2018 [18]. Created with biorender.com.

**Figure 2 biomedicines-12-00759-f002:**
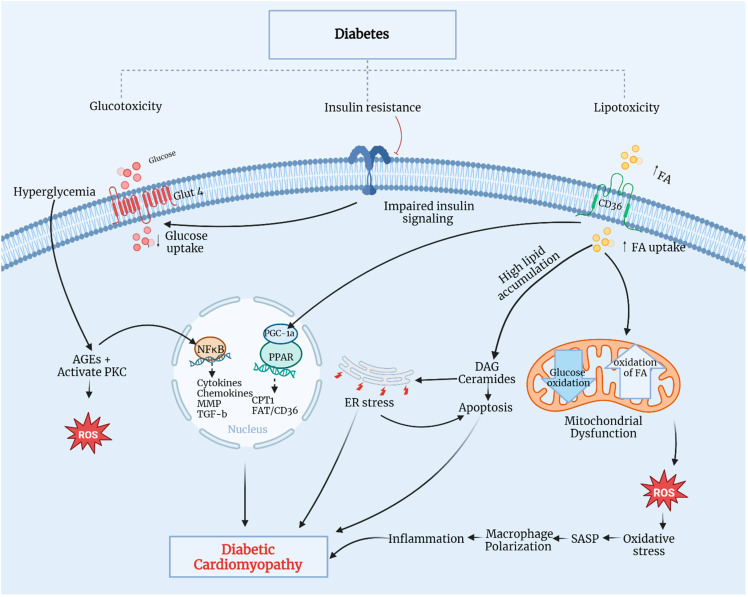
Pathophysiological mechanisms of DCM. CD36, a cluster of differentiation 36; CPT1, carnitine palmitoyl-transferase 1; DAG, diacylglycerol; ER, endoplasmic reticulum; FA, fatty acid; Glut4, glucose transporter 4; MMP, matrix metalloproteinase; NF-κB, nuclear factor kappa-B; PGC-1α, PPAR gamma cofactor-1α; PKC, protein kinase C; PPAR, peroxisome proliferator-activated receptor; ROS, reactive oxygen species; SASP, senescence-associated secretory phenotype; TGF-β, transforming growth factor-β. Modified from Sharma et al., 2022 [19]. Created with biorender.com.

## Data Availability

Not applicable.

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
