# Peer review of "Interplay between Senescence and Macrophages in Diabetic Cardiomyopathy: A Review of the Potential Role of GDF-15 and Klotho"

_biomedicines, 2024, doi:10.3390/biomedicines12040759_

Round 1
Reviewer 1 Report
Comments and Suggestions for Authors
The manuscript has its value in summarizing in vitro results. Considering the flood of publications with in vitro results this seems to be urgent, therefore I applaud the authors to undertake this task. Mostly the article succeedes in this task. I also strongly agree with the need of targeted studies in healthy and diabetic populations.
The article may be improved in some specific points. The title promises a close look at GDF-15 and Klotho and suggests a direct link. For my (limited) knowledge not every aspect has been included. Among the topics I'd like to have emphasized or corrected are:
-
the interplay between (micro-)vascular coronary damage, hypertonus and cardiac insufficiency/hypertrophy lacks the hypertension facvtor.
-
The authors at multuiple occasions mention hyperlipidemia which is a term with insufficient specificity. It summarizes free fatty acids, triglycerides hypercholesterolemia, high LD and HDL ... Dm affects these populations in different ways theresore specific terms should be used.
-
Similarly, acute and chronic inflammation is not separated. I am not sufficiently familiar with the primary studies to judge whether the experiments mimic one or the other type, bvut a clarification should be included.
-
In macrophages Klotho stimulates TNF-a which would be a proinflammatory cytokine. This is in contrast to the statement that in Dm Klotho is low but inflammation high.
-
Also, GDF-15 receptor regulation is described but not sufficiently proven (low receptor density at high GDF-15). The alternative explanation of low GDF-15 is not discussed.
-
In chapter 5 the authors mostly repeat the effects described in chapter 4.
Eliminationg these inconsistencies in my opinion would improve the article. Also, it might be improved if the lacking but necessary human data would be specified i.e. a rough study layout provided with the expected information and its relevance for Dm and DCM, Klotho and/or GDF-15.
Figure 1 is meaningless; it might be improved if a bit more specific actions of these factors are supplied within in the Figure, like: Insulin resistance --> [hyperglycemia --> hyperlipidemia, AGE --> inflammation] and --> [insulin resistance --> hyperinsulinemia] and --> [metabolic changes --> mitochondrial dysfunction] etc. I agree that there are multiple interplays between these lines but they may be adressed by arrows between lines. The end result would be cell senescence and apoptosis. Alternatively it might be fused with Fig. 2 (here, receöptors and hannels should be located within the membrane (since I am a visual reader like most others this slight oversight is irritating.
Author Response
Reviewer #1’s Comments and Suggestions
The manuscript has its value in summarizing in vitro results. Considering the flood of publications with in vitro results this seems to be urgent, therefore I applaud the authors to undertake this task. Mostly the article succeeds in this task. I also strongly agree with the need of targeted studies in healthy and diabetic populations.
The article may be improved in some specific points. The title promises a close look at GDF-15 and Klotho and suggests a direct link. For my (limited) knowledge not every aspect has been included. Among the topics I'd like to have emphasized or corrected are:
Response:
Thank you for your insights and suggestions to improve the manuscript. We greatly appreciate your feedback.
the interplay between (micro-)vascular coronary damage, hypertonus and cardiac insufficiency/hypertrophy lacks the hypertension factor.
Response:
Many thanks for your valuable comment. As requested, a paragraph explaining the interplay between microvascular coronary damage, hypertension, and diabetes was added (Page 5, lines 123–145) as follows:
[In diabetes mellitus, coronary microvascular dysfunction is an important characteristic of diabetes mellitus-related complications. It is linked to various other risk factors that significantly affect cardiovascular disease morbidity and mortality. Chronic uncontrolled high blood glucose contributes significantly to dysfunction of microcirculation as a common chronic feature. The pathogenesis of this microvascular complication is complex and not completely understood (1). However, hyperglycaemia contributes to the activation of various signaling pathways at microvascular levels, resulting in remodeling of the microvessels as well as myocardial tissue, with possible impairment of the microvascular supply to the myocardium (2). Microvascular remodeling includes thickening of the arteriole wall, narrowing of the lumen, fibrosis, and capillary rarefaction. Similarly, hypertension is another risk factor that leads to coronary microvascular alterations and promotes the progression of endothelial dysfunction and capillary rarefaction (3). Hypertension-induced changes in myocardial morphology and function also contribute to left ventricular hypertrophy and myocardial fibrosis (3); indeed, hypertension and type 2 diabetes are common comorbidities. Accordingly, uncontrolled blood glucose levels and blood pressure are interlinked because of diabetes-related or hypertension-related cardiovascular complications associated primarily with microvascular function deterioration. Both diseases can stimulate a mutual molecular mechanism, such as oxidative stress, inflammatory and immune system activation, and others. All of these are linked to microvascular dysfunctions that possibly contribute to the relationship between diabetes, hypertension, and coronary microvascular disorders (4).]
References:
(1) Salvatore T, Galiero R, Caturano A, Vetrano E, Loffredo G, Rinaldi L, Catalini C, Gjeloshi K, Albanese G, Di Martino A, Docimo G, Sardu C, Marfella R, Sasso FC. Coronary Microvascular Dysfunction in Diabetes Mellitus: Pathogenetic Mechanisms and Potential Therapeutic Options. Biomedicines. 2022 Sep 14;10(9):2274. doi: 10.3390/biomedicines10092274. PMID: 36140374; PMCID: PMC9496134.
(2) Sezer M, Kocaaga M, Aslanger E, Atici A, Demirkiran A, Bugra Z, Umman S, Umman B. Bimodal Pattern of Coronary Microvascular Involvement in Diabetes Mellitus. J Am Heart Assoc. 2016 Nov 14;5(11):e003995. doi: 10.1161/JAHA.116.003995. Erratum in: J Am Heart Assoc. 2017 Jan 11;6(1): PMID: 27930353; PMCID: PMC5210326.
(3) Zdravkovic M, Popadic V, Klasnja S, Klasnja A, Ivankovic T, Lasica R, Lovic D, Gostiljac D, Vasiljevic Z. Coronary Microvascular Dysfunction and Hypertension: A Bond More Important Than We Think. Medicina (Kaunas). 2023 Dec 11;59(12):2149. doi: 10.3390/medicina59122149. PMID: 38138252; PMCID: PMC10744540.
(4) Petrie JR, Guzik TJ, Touyz RM. Diabetes, Hypertension, and Cardiovascular Disease: Clinical Insights and Vascular Mechanisms. Can J Cardiol. 2018 May;34(5):575-584. doi: 10.1016/j.cjca.2017.12.005. Epub 2017 Dec 11. PMID: 29459239; PMCID: PMC5953551.
The authors at multiple occasions mention hyperlipidemia which is a term with insufficient specificity. It summarizes free fatty acids, triglycerides hypercholesterolemia, high LD and HDL ... Dm affects these populations in different ways theresore specific terms should be used.
Response:
Many thanks for your comment. As requested, a paragraph has been added to provide more specific information regarding the effect of lipids on M1 and M2 polarization (Page 8: lines 204–215) as follows:
[Lipids have a substantial effect on macrophage polarization and influence macrophage functions. Previous studies have shown that oxidized low-density lipoproteins (oxLDLs) induced M1 polarization, thereby contributing to the inflammatory process of atherosclerosis (1, 2). For example, oxLDL induced the production of pro-inflammatory cytokines, the expression of HLA-DR and CD86, and T cell proliferation (3). Another study reported that the exposure of monocytes to a low concentration of oxLDL induced epigenetic histone modifications that resulted in a long-lasting proatherogenic macrophage phenotype characterized by increased proinflammatory cytokine production and foam cell formation (4). Peroxisome-proliferator–activated receptors (PPARs) and liver X receptors (LXRs) are other well-known lipid-binding factors with known effects on macrophage polarization (5).]
- Pireaux V, Sauvage A, Bihin B, Van Steenbrugge M, Rousseau A, Van Antwerpen P, Zouaoui Boudjeltia K, Raes M. Myeloperoxidase-Oxidized LDLs Enhance an Anti-Inflammatory M2 and Antioxidant Phenotype in Murine Macrophages. Mediators Inflamm. 2016;2016:8249476 doi: 10.1155/2016/8249476
- de la Paz Sánchez-Martínez M, Blanco-Favela F, Mora-Ruiz MD, Chávez-Rueda AK, Bernabe-García M, Chávez-Sánchez L. IL-17-differentiated Macrophages Secrete Pro-Inflammatory Cytokines in Response to Oxidized Low-Density Lipoprotein. Lipids Health Dis. 2017 Oct 10;16(1):196. doi: 10.1186/s12944-017-0588-1. PMID: 29017604; PMCID: PMC5634956.
- Chávez-Sánchez L, Garza-Reyes MG, Espinosa-Luna JE, Chávez-Rueda K, Legorreta-Haquet MV, Blanco-Favela F. The Role of TLR2, TLR4 and CD36 in Macrophage Activation and Foam Cell Formation in Response to oxLDL in Humans. Hum Immunol. 2014 Apr;75(4):322-329. doi: 10.1016/j.humimm.2014.01.012. Epub 2014 Jan 30. PMID: 24486576.
- Bekkering S, Quintin J, Joosten LA, van der Meer JW, Netea MG, Riksen NP. Oxidized Low-Density Lipoprotein Induces Long-Term Proinflammatory Cytokine Production and Foam Cell Formation via Epigenetic Reprogramming of Monocytes. Arterioscler Thromb Vasc Biol. 2014 Aug;34(8):1731-1738. doi: 10.1161/ATVBAHA.114.303887. Epub 2014 Jun 5. PMID: 24903093.
- Zizzo G, Cohen PL. The PPAR-γ Antagonist GW9662 Elicits Differentiation of M2c-like Cells and Upregulation of the MerTK/Gas6 Axis: a Key Role for PPAR-γ in Human Macrophage Polarization. J Inflamm (Lond). 2015 May 3;12:36. doi: 10.1186/s12950-015-0081-4. PMID: 25972766; PMCID: PMC4429687.
Similarly, acute and chronic inflammation is not separated. I am not sufficiently familiar with the primary studies to judge whether the experiments mimic one or the other type, but a clarification should be included.
Response:
We highly appreciate your constructive comment. We clarified this by adding new material in the manuscript (Pages 7, 8, lines 159–174) as follows:
“Inflammatory processes significantly contribute to the pathogenesis of DCM (1). In particular, T2DM induces chronic low-grade inflammation in the heart tissue, leading to severe structural and functional alterations (2), while elevated glucose levels cause the release of proinflammatory cytokines (interleukin [IL]-6, IL-18, pro-IL-1ß, and tumor necrosis factor-alpha [TNF-α]) (3–5). Recent studies have demonstrated fibrosis and tissue damage, which are both features of chronic inflammation, in an STZ-induced T2DM animal model (2,6). Moreover, both in vitro and animal studies have implicated elevated blood glucose levels as substantial contributors to diabetes-induced myocardial fibrosis. For example, the cultivation of cardiac fibroblasts in a medium containing a high concentration of glucose causes the cells to produce excessive quantities of ECM proteins, including collagen, fibronectin, and matricellular macromolecules (7,8). The development of diabetic cardiomyopathy in the STZ-induced T2DM animal model was also associated with an increase in cardiac immune cell invasion, specifically T lymphocytes and macrophages, indicating chronic cardiac inflammation (9–12). These findings suggest a chronic inflammatory condition in the myocardial interstitium that can potentially lead to cardiac dysfunction and an increased risk of cardiac events such as DCM.”
References:
- Frati G, Schirone L, Chimenti I, Yee D, Biondi-Zoccai G, Volpe M, et al. An Overview of the Inflammatory Signalling Mechanisms in the Myocardium Underlying the Development of Diabetic Cardiomyopathy. Cardiovas Res 2017;113(4):378–388. doi:10.1093/cvr/cvx011.
- Tsai T-H, Lin C-J, Chua S, Chung S-Y, Chen S-M, Lee C-H, et al. Deletion of RasGRF1 Attenuated Interstitial Fibrosis in Streptozotocin-Induced Diabetic Cardiomyopathy in Mice through Affecting Inflammation and Oxidative Stress. Int J Mol Sci. 2018, 19(10), 3094. doi:10.3390/ijms19103094.
- Jia, G.; Whaley-Connell, A.; Sowers, J. R. Diabetic Cardiomyopathy: A Hyperglycaemia- and Insulin-Resistance-Induced Heart Disease. Diabetologia 2018; 61(1): 21–28. doi:10.1007/s00125-017-4390-4.
- Li, J.; Ma, W.; Yue, G.; Tang, Y.; Kim, I.; Weintraub, N. L.; et al. Cardiac Proteasome Functional Insufficiency Plays a Pathogenic Role in Diabetic Cardiomyopathy. J Mol Cell Cardiol. 2017;102:53-60. doi:10.1016/j.yjmcc.2016.11.013.
- Parim B, Sathibabu Uddandrao VV, Saravanan G. Diabetic Cardiomyopathy: Molecular Mechanisms, Detrimental Effects of Conventional Treatment, and Beneficial Effects of Natural Therapy. Heart Fail Rev. 2019;24(2):279-299. doi:10.1007/s10741-018-9749-1.
- Udumula MP, Mangali S, Kalra J, Dasari D, Goyal S, Krishna V, et al. High Fructose and Streptozotocin Induced Diabetic Impairments Are Mitigated By Indirubin-3-Hydrazone via Downregulation of PKR Pathway in Wistar Rats. Sci Rep. 2021;11(1):12924. doi:10.1038/s41598-021-92345-2.
- Muona P, Peltonen J, Jaakkola S, Uitto J. Increased Matrix Gene Expression by Glucose in Rat Neural Connective Tissue Cells in Culture. Diabetes. 1991;40(5):605-611. doi:10.2337/diab.40.5.605.
- Singh VP, Baker KM, Kumar R. Activation of the Intracellular Renin-Angiotensin System in Cardiac Fibroblasts by High Glucose: Role In Extracellular Matrix Production. Am J Physiol Heart Circ Physiol 2008;294(4):H1675-H1684. doi:10.1152/ajpheart.91493.2007.
- Bugger H, Abel ED. Molecular Mechanisms of Diabetic Cardiomyopathy. Diabetologia. 2014;57(4):660-671. doi:10.1007/s00125-014-3171-6.
- Becher PM, Lindner D, Fröhlich M, Savvatis K, Westermann D, Tschöpe C. Assessment of Cardiac Inflammation And Remodeling During the Development of Streptozotocin-Induced Diabetic Cardiomyopathy In Vivo: A Time Course Analysis. Int J Mol Med 2013;32(1):158-164. doi:10.3892/ijmm.2013.1368.
- Hu X, Bai T, Xu Z, Liu Q, Zheng Y, Cai L. Pathophysiological Fundamentals of Diabetic Cardiomyopathy. In Comprehensive Physiology; Wiley, 2017; pp 693–711. doi:10.1002/cphy.c160021.
- Kanamori H, Naruse G, Yoshida A, Minatoguchi S, Watanabe T, Kawaguchi T, et al. Morphological Characteristics in Diabetic Cardiomyopathy Associated with Autophagy. J Cardiol. 2021;77(1):30-40. doi:10.1016/j.jjcc.2020.05.009.
In macrophages Klotho stimulates TNF-a which would be a proinflammatory cytokine. This is in contrast to the statement that in Dm Klotho is low but inflammation high.
Response:
Thank you for raising this speculation. To our knowledge, no direct evidence suggests that macrophages stimulate TNF-α production through the action of klotho. Although research specifically investigating the relationship between macrophages, klotho, and TNF-α is limited in the context of diabetes, current evidence suggests that klotho mediates anti-inflammatory effects by inhibiting the production of proinflammatory cytokines, including TNF-α. Therefore, klotho may protect the diabetic heart from structural deterioration and loss of functional integrity. In addition, studies in various disease models have shown that klotho deficiency is associated with increased inflammation and immune dysfunction [1–4].
References:
- Zhao Y, et al. Klotho Depletion Contributes to Increased Inflammation in Kidney of the db/db Mouse Model Of Diabetes Via RelA (serine)536 Phosphorylation. Diabetes. 2011;60(7):1907-1916.
- Fitzpatrick EA, et al. Role of Fibroblast Growth Factor-23 in Innate Immune Responses. Front Endocrinol (Lausanne). 2018;9:320.
- Zhou X, Lei H, Sun Z. Participation of Immune Cells in Klotho Deficiency-induced Salt-sensitive Hypertension. FASEB J. 2015;29(S1):667.3.
- Witkowski JM, et al., Klotho–A Common Link in Physiological and Rheumatoid Arthritis-Related Aging of Human CD4+ Lymphocytes. J Immunol. 2007;178(2):771-777.
Accordingly, we highlighted the statements in the manuscript indicating that klotho overexpression exerts anti-inflammatory effects by reducing proinflammatory cytokines, such as IL-6 and TNF-α (lines 293–295, Page 10); (lines 343–345, Page 11).
Also, GDF-15 receptor regulation is described but not sufficiently proven (low receptor density at high GDF-15). The alternative explanation of low GDF-15 is not discussed.
Response:
Thank you for your comment. We have added this missing information (lines 379–391, Page 12), as follows:
“Glial-derived neurotrophic factor family receptor α-like (GFRAL) is the receptor for GDF-15, which is only detected in the brain. Upon binding to GFRAL, GDF-15 promotes the activation of the coreceptor tyrosine kinase RET [1, 2]. This GDF-15/GFRAL/RET complex plays a critical role in weight regulation, as indicated by the promotion of weight loss in obese mice by recombinant GDF-15, which triggers a reduction in food intake. This anti-obesity action of GDF-15 is abrogated in GFRAL gene-deleted mice [2], while diet-induced obesity is exacerbated in GFRAL-deficient mice [2]. Remarkably, transgenic mice that overexpress GDF-15 exhibit enhanced insulin sensitivity and glucose tolerance along with a marked reduction in body weight [3, 4]. These beneficial metabolic effects provide new insights into GDF-15 as a potential target for diabetes treatment. Moreover, GDF-15 expression has a protective effect, as the absence of GDF15 is associated with increased damage in various tissues. For example, GDF-15 knockout mice demonstrated cardiac rupture after myocardial infarction [5].”
References:
- Yang L, Chang C-C, Sun Z, et al. GFRAL is the Receptor for GDF15 and is Required for the Anti-obesity Effects of the Ligand. Nat Med. 2017;23(10):1158-1166.
- Mullican SE, Lin-Schmidt X, Chin C-N, et al. GFRAL is the Receptor for GDF15 and the Ligand Promotes Weight Loss in Mice and Nonhuman Primates. Nat Med. 2017;23(10):1150-1157.
- Macia L, Tsai VW-W, Nguyen AD, et al. Macrophage Inhibitory Cytokine 1 (MIC-1/GDF15) Decreases Food Intake, Body Weight and Improves Glucose Tolerance in Mice on Normal & Obesogenic Diets. PloS One 2012;7(4):e34868.
- Tsai V, Zhang H, Manandhar R, et al. Treatment with the TGF-b Superfamily Cytokine MIC-1/GDF15 Reduces the Adiposity and Corrects the Metabolic Dysfunction of Mice with Diet-induced Obesity. Int J Obes (Lond). 2018; 42(3):561-571.
- Kempf T, Zarbock A, Widera C, et al. GDF-15 is an Inhibitor of Leukocyte Integrin Activation Required For Survival After Myocardial Infarction in Mice. Nat Med. 2011;17(5):581-588.
In chapter 5 the authors mostly repeat the effects described in chapter 4.
Eliminationg these inconsistencies in my opinion would improve the article. Also, it might be improved if the lacking but necessary human data would be specified i.e. a rough study layout provided with the expected information and its relevance for Dm and DCM, Klotho and/or GDF-15.
Response:
Thank you for your comment. We have addressed this by eliminating the inconsistencies, adding more evidence to support this chapter, and highlighting human-based studies. (lines 273–276, Page 9); (lines 283–319, Page 10); (lines 321–331, Pages 10, 11)
“The interplay between senescence and macrophages is a critical factor underlying DCM progression. Under hyperglycemic conditions, cardiomyocytes in diabetic patients undergo stress-induced premature senescence. The senescent cardiomyocytes release inflammatory cytokines that recruit monocytes, which differentiate into macrophages and infiltrate the myocardium. The senescent cardiomyocytes can directly modulate macrophage polarization toward the M1 phenotype by releasing SASP components. These macrophages sustain the senescent phenotype by amplifying the pro-inflammatory response [1]. Notably, macrophages were shown to regulate cellular senescence by influencing their microenvironment in a p16INK4a/Luc mouse model [2].
M1 macrophages promote senescence by secreting inflammatory cytokines, such as TNF-α and IL-6 [1, 3]. The presence of senescent cells and macrophages further contributes to cardiac remodeling, fibrosis, and impaired contractility. The interplay between senescence and macrophages facilitates cardiac fibrosis, as the senescent cells produce MMPs that degrade the ECM, while macrophages enhance the deposition of fibrotic tissue [1]. In support of this interplay, chemotherapy in breast cancer cells with a mutated TP53 gene promotes senescent cells and cellular survival via phagocytosis and macrophages, respectively [4], indicating a functional correlation between senescence and macrophages.
In a streptozotocin-diabetic mouse model, the pathological features of cardiomyopathy were attenuated by the in vivo administration of Klotho for 12 weeks. Klotho was found to reduce inflammatory and stress-related features in H9C2 cardiomyoblasts exposed to high levels of glucose [5]. In individuals with acute heart failure, the soluble alpha-Klotho circulatory level was significantly correlated with patients’ responsiveness to intensive treatment [6]. In line with this, the genetic mutation of Klotho accelerates aging in mice in different physiological systems and organs, including cardiac muscle [7]. A previous review of the preanalytical characteristics of GDF-15 in clinical settings indicated that a reduced level of circulatory GDF-15 is linked to longevity [8]. In support of this, the level of GDF-15 in aged and healthy individuals was correlated with age. Further, in chronic heart failure patients, the GDF15 serum level was elevated and associated with the severity of the disease [9].
In cardiomyocytes, functional alterations of replicative senescence are associated with chronic inflammation and cell death [10]. A study of 20 heart failure patients revealed that inflammation increased the amounts of extracellular matrix in the endomyocardium and affected cardiac remodeling [11]. A proteomic study of serum samples isolated from cardiomyopathy patients showed an association between dysregulation in extracellular matrix protein and DCM [12]. Pathological changes in myocardial remodeling were also modulated by senescent cardiomyocytes [13], while genetic profiling of macrophages was correlated with changes associated with myocardial infarction healing and modeling [14]. Immunohistochemical studies conducted on biopsy samples isolated from DCM patients have shown significant immunoreactivity for macrophage markers, including those associated with whole and M2 macrophages [15]. This evidence supports the occurrence of cross-talk between senescence and macrophages and the contribution of macrophages to DCM and cardiac remodeling as we age.”
References:
- Elder SS, Emmerson E. Senescent Cells and Macrophages: Key Players for Regeneration? Open Biol. 2020;10(12):200309.
- Hall BM, et al. Aging of Mice Is Associated With p16(Ink4a)- and Β-Galactosidase-Positive Macrophage Accumulation That Can Be Induced in Young Mice by Senescent Cells. Aging (Albany NY). 2016;8(7):1294-315.
- Henson SM, Aksentijevic D. Senescence and Type 2 Diabetic Cardiomyopathy: How Young Can You Die of Old Age? Front Pharmacol. 2021;12:716517.
- Tonnessen-Murray CA, et al. Chemotherapy-induced Senescent Cancer Cells Engulf Other Cells to Enhance Their Survival. J Cell Biol. 2019;218(11):3827-3844.
- Li X, et al. Klotho Improves Diabetic Cardiomyopathy by Suppressing the Nlrp3 Inflammasome Pathway. Life Sci. 2019;234:116773.
- Taneike M, et al. Alpha-Klotho is a Novel Predictor of Treatment Responsiveness in Patients with Heart Failure. Sci Rep. 2021;11(1):2058.
- Kuro-o M, et al. Mutation of the Mouse Klotho Gene Leads to a Syndrome Resembling Ageing. Nature. 1997;390(6655):45-51.
- Wollert KC, Kempf T, Wallentin L. Growth Differentiation Factor 15 as a Biomarker in Cardiovascular Disease. Clin Chem. 2017;63(1):140-151.
- Kempf T, et al. Circulating Concentrations of Growth-Differentiation Factor 15 in Apparently Healthy Elderly Individuals and Patients with Chronic Heart Failure as Assessed by a New Immunoradiometric Sandwich Assay. Clin Chem. 2007;53(2):284-291.
- Gude NA, et al. Cardiac Ageing: Extrinsic and Intrinsic Factors in Cellular Renewal and Senescence. Nat Rev Cardiol. 2018;15(9):523-542.
- Westermann D, et al. Cardiac Inflammation Contributes to Changes in the Extracellular Matrix in Patients With Heart Failure and Normal Ejection Fraction. Circulation: Heart Fail. 2011;4(1):44-52.
- Klimentova J, et al. Proteomic Profiling of Dilated Cardiomyopathy Plasma Samples ─ Searching for Biomarkers with Potential to Predict the Outcome of Therapy. J Proteome Res. 2024;23(3):971-984.
- Redgrave RE, et al. Senescent Cardiomyocytes Contribute To Cardiac Dysfunction Following Myocardial Infarction. npj Aging. 2023;9(1):15.
- Mouton AJ, et al. Mapping Macrophage Polarization Over the Myocardial Infarction Time Continuum. Basic Res Cardiol. 2018;113(4):26.
- Nakayama T, et al. Clinical Impact of the Presence of Macrophages in Endomyocardial Biopsies of Patients with Dilated Cardiomyopathy. Eur J Heart Fail. 2017;19(4):490-498.
Figure 1 is meaningless; it might be improved if a bit more specific actions of these factors are supplied within in the Figure, like: Insulin resistance --> [hyperglycemia --> hyperlipidemia, AGE --> inflammation] and --> [insulin resistance --> hyperinsulinemia] and --> [metabolic changes --> mitochondrial dysfunction] etc. I agree that there are multiple interplays between these lines but they may be adressed by arrows between lines. The end result would be cell senescence and apoptosis. Alternatively it might be fused with Fig. 2 (here, receöptors and hannels should be located within the membrane (since I am a visual reader like most others this slight oversight is irritating.
Response:
Thank you for this valuable suggestion. The figures (Figs. 1 & 2) have been modified and improved accordingly (pages 3 & 6).

Reviewer 2 Report
Comments and Suggestions for Authors
In the manuscript #biomedicines-2910036 by Almohaimeed et al., " Interplay between Senescence and Macrophages in Diabetic Cardiomyopathy: A Review of the Potential Role of GDF-15 and Klotho ", authors attempt to review the role of GDF-15 and Klotho in the interaction between aging and macrophages in diabetic type 2 cardiomyopathy. They cite only reference #59 for Klotho in diabetic cardiac hypertrophy, and only cite another review of reference #63 for GDF-15. It is scientifically unsound and does not contain sufficient interest and originality to merit publication. Furthermore, Figure 1 shows a scattering of spelling errors., and Figure 2 shows the receptor outside the cell, which is not an appropriate diagram.
Comments on the Quality of English LanguageExtensive editing of English language required.
Author Response
Reviewer #2’s Comments and Suggestions
In the manuscript #biomedicines-2910036 by Almohaimeed et al., " Interplay between Senescence and Macrophages in Diabetic Cardiomyopathy: A Review of the Potential Role of GDF-15 and Klotho ", authors attempt to review the role of GDF-15 and Klotho in the interaction between aging and macrophages in diabetic type 2 cardiomyopathy. They cite only reference #59 for Klotho in diabetic cardiac hypertrophy, and only cite another review of reference #63 for GDF-15. It is scientifically unsound and does not contain sufficient interest and originality to merit publication. Furthermore, Figure 1 shows a scattering of spelling errors., and Figure 2 shows the receptor outside the cell, which is not an appropriate diagram.
Response:
Thank you for this valuable suggestion and feedback. We have comprehensively addressed the issue raised regarding klotho and GDF-15 in diabetic cardiac hypertrophy, and the relevant figures (Fig. 1 & 2) have been modified and improved accordingly (Pages 3 & 6). The following information has been added according to your suggestion as follows:
(page 11, lines 354–364)
“A recent study by Donate-Correa et al. reported reduced Klotho levels in adults with T2DM and preserved kidney function. Similarly, reduced Klotho levels have been associated with increased levels of inflammatory markers and a higher incidence of vascular disease and subclinical atherosclerosis (1). These findings suggest that Klotho may be involved in the development and progression of diabetes-related vascular complications. Low levels of Klotho have also been found in adults with a high risk of cardiovascular disease, such as obesity, smoking, diabetes, and high levels of total cholesterol and triglycerides (2). In both in vitro and in vivo experiments, Klotho treatment effectively suppressed high glucose–induced inflammation, ROS generation, and cardiac cell death, leading to improved cardiac function (3).”
(pages 12 & 13, lines 399–402)
“It is known that GDF-15 is induced in response to conditions that promote hypertrophy and dilated cardiomyopathy. Interestingly, transgenic mice with cardiac-specific overexpression of GDF15 have shown partial resistance to pressure overload–induced hypertrophy (4).”
(page 13, lines 408–413)
“Furthermore, GDF-15 levels have been positively associated with markers of poor glycemic control, such as high HbA1c levels (5–8). Additionally, elevated GDF-15 levels have been associated with an increased risk of diabetic complications, such as DCM, chronic kidney disease, and diabetic retinopathy (9,10,11). Jurczyluk et al. found that GDF-15 levels are greatly increased in cardiomyocytes following an ischemic event and post-reperfusion (12).”
References:
- Donate-Correa J, Martín-Núñez E, Mora-Fernández C, González-Luis A, Martín-Olivera A, Navarro-González JF. Association of Klotho with Coronary Artery Disease in Subjects with Type 2 Diabetes Mellitus and Preserved Kidney Function: A Case-Control Study. Int J Mol Sci. 2023; 24(17):13456. doi:10.3390/ijms241713456.
- Lee J, Kim D, Lee H, Choi J.-Y, Min J.-Y, Min K-B. Association Between Serum Klotho Levels and Cardiovascular Disease Risk Factors in Older Adults. BMC Cardiovasc Disord. 2022; 22(1):442. doi:10.1186/s12872-022-02885-2.
- Guo Y, Zhuang X, Huang Z, Zou J, Yang D, Hu X, et al. Klotho Protects the Heart From Hyperglycemia-Induced Injury by Inactivating ROS and NF-κB-mediated Inflammation Both in Vitro and in Vivo. BBA-Mol Basis Dis. 2018;1864(1):238-251. doi:10.1016/j.bbadis.2017.09.029.
- Wollert KC, Kempf T, Wallentin L. Growth Differentiation Factor 15 as a Biomarker in Cardiovascular Disease. Clin Chem. 2017;63(1):140-151. doi:10.1373/clinchem.2016.255174.
- Adela R, Banerjee SK. GDF-15 as a Target and Biomarker for Diabetes and Cardiovascular Diseases: A Translational Prospective. J Diabet Res. 2015; 2015:1-14. doi:10.1155/2015/490842.
- Echouffo-Tcheugui JB, Daya N, Ndumele CE, Matsushita K, Hoogeveen RC, Ballantyne CM, et al. Diabetes, GDF-15 and Incident Heart Failure: The Atherosclerosis Risk in Communities Study. Diabetologia. 2022;65(6):955-963. doi:10.1007/s00125-022-05678-6.
- Lindahl B. The Story of Growth Differentiation Factor 15: Another Piece of the Puzzle. Clin Chem. 2013;59(11):1550-1552. doi:10.1373/clinchem.2013.212811.
- Ding Q, Mracek T, Gonzalez-Muniesa P, Kos K, Wilding J, Trayhurn P, et al. Identification of Macrophage Inhibitory Cytokine-1 in Adipose Tissue and Its Secretion as an Adipokine by Human Adipocytes. Endocrinology 2009;150(4):1688-1696. doi:10.1210/en.2008-0952.
- Hoshide S, Kario K. Elevated Growth And Differentiation Factor 15 (Gdf-15) Levels Amplifies The Association Between Home Blood Pressure Variability And Cardiovascular Outcome. J Hypertens. 2023;41(Suppl 3):e131. doi:10.1097/01.hjh.0000940108.15249.ad.
- Jurczyluk J, Brown D, Stanley KK. Polarised Secretion of Cytokines in Primary Human Microvascular Endothelial Cells is not Dependent on N ‐linked Glycosylation. Cell Biol Int 2003;27(12):997-1003. doi:10.1016/j.cellbi.2003.09.002.
Extensive editing of English language required.
Response:
Thank you for your comment and suggestion. We had this manuscript edited using the Elsevier Language Editing Services before submitting it to Biomedicines (the English Editing Certificate is attached at the end of attached file). However, if you still believe that this manuscript requires further English language editing, we do not mind doing this; however, we would prefer to delay this editing process until after approval of the revisions by the reviewers (after amendments), so that the English editing is the last step in the manuscript publication process.

Reviewer 3 Report
Comments and Suggestions for Authors
It is fascinating review concerning the interplay between senescence and macrophages in diabetic cardiomyopathy, a review of the potential role of GDF-15 and klotho protein.
I have only few suggestions:
1. In the abstract the shortcut GDF-15 should be explained.
2. It is worth to mention in the same line that Klotho is a protein
3. Line 66 it is not clear what is advanced glycation end product (AGEs)
4. Line 291: I doubt the sentence: „by GDF-15 overexpression in GDF15-deficient mice”. Is it a mistake?
There is not a lot of information about the connection between GDF-15 and Klotho protein in diabetic cardiomyopathy.
Comments on the Quality of English LanguageMinor editing of English language required
Author Response
Reviewer #3’s Comments and Suggestions
It is fascinating review concerning the interplay between senescence and macrophages in diabetic cardiomyopathy, a review of the potential role of GDF-15 and klotho protein.
Response:
We sincerely appreciate your insights and suggestions, as they have dramatically improved the manuscript. Thank you for your valuable feedback.
I have only few suggestions:
- In the abstract the shortcut GDF-15 should be explained.
Response:
Thank you for your comment. We have addressed this in the abstract (lines 17–24, Page 1)
“Evidence indicates that growth differentiation factor-15 (GDF-15), a protein that belongs to the transforming growth factor-beta (TGF-β) superfamily, is associated with age-related diseases and exerts an anti-inflammatory role in various disease models. Although some evidence suggests that GDF-15 can preserve klotho, a transmembrane antiaging protein, emerging research has elucidated the potential involvement of GDF-15 and klotho in the interplay that occurs between macrophage-induced inflammation and cellular senescence in the context of DCM.”
- It is worth to mention in the same line that Klotho is a protein
Response:
Thank you for your comment. We have addressed this in the abstract (line 20, Page 1).
“klotho, a transmembrane antiaging protein,…”
- Line 66 it is not clear what is advanced glycation end product (AGEs)
Response:
Thank you for bringing this to our attention. We have clarified this in the manuscript (Lines 72–74, Page 2) as:
“Increased production and accumulation of AGEs in cardiac cells as a consequence of prolonged hyperglycemia…”
In addition, the interplays between the various etiological factors, including AGEs, that contribute to the development of diabetic cardiomyopathy are now depicted in the modified Figure 1 (Page 3) and indicated by arrows for further clarification of these relationships. In addition, the role of AGEs as contributing factors to the development of diabetic cardiomyopathy has been explained in the manuscript (lines 111–119, Pages 4 & 5) as follows:
“Prolonged hyperglycemia promotes AGE formation and protein kinase-C activation, thereby initiating oxidative stress by stimulating ROS production and ultimately causing oxidative damage to cardiomyocytes and impaired myocardial contractility [1, 2]. A significant increase in AGEs triggers nuclear factor kappa-B (NF-κB) signaling pathways that induce inflammation, cytokine and chemokine production, and activation of profibrotic factors, such as matrix metalloproteinase (MMP) and transforming growth factor-beta (TGF-β), as well as myocyte apoptosis. These pathways can cause functional and structural damage that leads to cardiomyocyte death, left ventricular remodeling, and systolic dysfunction [3].”
References:
- Jia G, Whaley-Connell A, Sowers JR. Diabetic Cardiomyopathy: A Hyperglycaemia- and Insulin-Resistance-Induced Heart Disease. Diabetologia. 2018;61(1):21-28. doi:10.1007/s00125-017-4390-4.
- Ritchie RH, Zerenturk EJ, Prakoso D, Calkin AC. Lipid Metabolism and its Implications for Type 1 Diabetes-Associated Cardiomyopathy. J Mol Endocrinol. 2017;58(4):R225-R240. doi:10.1530/JME-16-0249.
- Salvatore T, Pafundi PC, Galiero R, Albanese G, Di Martino A, Caturano A, et al. The Diabetic Cardiomyopathy: The Contributing Pathophysiological Mechanisms. Front Med. 2021;8:695792. doi:10.3389/fmed.2021.695792.
4.Line 291: I doubt the sentence: „by GDF-15 overexpression in GDF15-deficient mice”. Is it a mistake?
Response:
Thank you for highlighting this issue. This was the result of an editing error. The sentence has been reviewed and corrected. (lines 434–437, Page 13)
There is not a lot of information about the connection between GDF-15 and Klotho protein in diabetic cardiomyopathy.
Response:
Thank you for your comment. Accordingly, we have added further details (page 13, lines 417–434) as follows:
“Currently, few studies have highlighted the link between GDF-15 and the Klotho protein in the context of DCM. Among different disease models, only one study that investigated the correlation between GDF-15 and Klotho protein in acute kidney injury and kidney fibrosis actually found that GDF-15 and Klotho protein levels show a strong correlation. GDF-15 enhanced Klotho expression in healthy mice and cultured tubular cells (1), whereas Klotho expression was reduced in GDF-15–deficient mice but was conserved after GDF-15 administration (23). Furthermore, GDF-15 and the Klotho protein are both involved in the development and progression of fibrosis, a hallmark of DCM (2,3). One study on idiopathic pulmonary fibrosis revealed increases in GDF-15 expression and accumulation in the extracellular matrix, suggesting that the increased expression and accumulation of GDF-15 in the extracellular matrix contributed to the fibrotic process associated with idiopathic pulmonary fibrosis. This finding highlights the potential role of GDF-15 in promoting fibrosis in the lungs (3). Another study using a mouse model of myocardial infarction reported that treatment with Klotho improved cardiac function and reduced cardiac fibrosis (4). Another investigation using a long-term rat model resembling type 1 diabetes mellitus found that reduced serum levels of Klotho in diabetic rats may promote the fibrotic process, suggesting a role for Klotho in the development of cardiac fibrosis (5).”
References:
1.Valiño-Rivas L, Cuarental L, Ceballos MI, Pintor-Chocano A, Perez-Gomez MV, Sanz AB, et al. Growth Differentiation Factor-15 Preserves Klotho Expression in Acute Kidney Injury and Kidney Fibrosis. Kidney Int. 2022;101(6):1200-1215. doi:10.1016/j.kint.2022.02.028.
- Mencke R, Olauson H, Hillebrands J-L. Effects of Klotho on Fibrosis and Cancer: A Renal Focus on Mechanisms and Therapeutic Strategies. Adv Drug Deliv Rev. 2017;121:85-100. doi:10.1016/j.addr.2017.07.009.
- Radwanska A, Cottage CT, Piras A, Overed-Sayer C, Sihlbom C, Budida R, et al. Increased Expression and Accumulation of Gdf15 in IPF Extracellular Matrix Contribute to Fibrosis. JCI Insight. 2022;7(16):E153058. doi:10.1172/jci.insight.153058.
- Wang K, Li Z, Ding Y, Liu Z, Li Y, Liu X, et al. Klotho Improves Cardiac Fibrosis, Inflammatory Cytokines, Ferroptosis, and Oxidative Stress in Mice With Myocardial Infarction. J Physiol Biochem. 2023;79(2):341-353. doi:10.1007/s13105-023-00945-5.
- Martín-Carro B, Martín-Vírgala J, Fernández-Villabrille S, Fernández-Fernández A, Pérez-Basterrechea M, Navarro-González JF, et al. Role of Klotho and AGE/RAGE-Wnt/β-Catenin Signalling Pathway on the Development of Cardiac and Renal Fibrosis in Diabetes. Int J Mol Sci 2023;24(6):5241. doi:10.3390/ijms24065241.
Minor editing of English language required.
Response:
Thank you for your comment and suggestion. We had this manuscript edited using the Elsevier Language Editing Services before submitting it to Biomedicines (the English Editing Certificate is attached at the end of the attached file). However, if you still believe that this manuscript requires further English language editing, we do not mind doing this; however, we would prefer to delay this editing process until after approval of the revisions by the reviewers (after amendments) so that the English editing is the last step in the manuscript publication process.

Round 2
Reviewer 2 Report
Comments and Suggestions for Authors
Referenced articles are now more plentiful, properly reviewed and the text has been improved.